# International Survey of the Usage of Whole Building Life Cycle Assessment Software

Thais Sartori *, Robin Drogemuller , Sara Omrani  and Fiona Lamari

School of Architecture and Built Environment, Faculty of Engineering, Queensland University of Technology (QUT), Brisbane City, QLD 4000, Australia
* Correspondence: thais.goncalvessartori@hdr.qut.edu.au

**Abstract:** The pressure to reduce the environmental impacts of buildings over their lifetime has driven certification bodies and the government to require a Life Cycle Assessment (LCA). However, LCA is a data-intensive and time-consuming process which complicates design activities, especially when performing a Whole Building LCA (WBLCA). Software tools can simplify the assessment by providing information more aligned with the users' needs. This research surveyed 178 building designers who utilise WBLCA software as a decision-making tool. The aim was to identify patterns in the usage of the software and provide guidance to WBLCA software developers. For this purpose, statistical analyses identified the software preferences within each group of users, e.g., the users' geographical location, professional background and years of WBLCA experience, among others. The results identified challenges faced by the construction industry, such as the need for more efficient communication among stakeholders. Therefore, attributes that allow designers to share information were rated as the most valuable. Two main groups of users were identified, and guidelines were drawn based on the profiles of the groups. Improving software support to designers will enable WBLCA to be integrated more efficiently with BPP by improving the users' experience and their ability to make more informed decisions.

**Keywords:** design; decision-making; life cycle assessment (LCA); whole building life cycle assessment; WBLCA; survey the usage of the software



## 1. Introduction

The construction industry significantly impacts the environment by consuming a considerable proportion of energy and natural resources. Buildings consume approximately 36% of primary energy and are responsible for 40% of energy-related greenhouse gas emissions [1]. Consequently, designers are challenged to drastically reduce the operational energy demand and embodied carbon of buildings over the next 30 years [2].

Life cycle assessment (LCA) is a standardised methodology that evaluates the environmental impacts of all system flows throughout a product's life cycle [3]. It is applied to products and processes, and it was first used in the construction sector in the 1980s [4]. Due to the increasing call for a life-cycle thinking approach, LCA credits in green building rating systems have been increasing (GBRS) [5–7], and codes and legislations are mandating LCA as part of the building permit requirements [8–11].

The standard for LCA of buildings [12] is very similar to the general LCA framework [3], which overloads design activities with data-intensive and time-consuming procedures. Software tools can facilitate this process by providing an interface more adapted to the Building Project Process (BPP) [13]. However, most of the existing environmental impact assessment tools do not adequately consider the designer's needs [14,15], particularly when analysing the assessment progress across the stages of BPP. For instance, Bleil de Souza [16] mentioned the inappropriateness of the input interface, which requires detailed information not always available early in design. Indeed, LCA tools are typically developed

by LCA experts, making them more appropriate for academic study or the development of Environmental Product Declaration (EPD) [17] rather than exploratory tools suited for early design. While scientists tend to focus on the underlying methods, designers are more concerned with achieving the desired result. In other words, while researchers are problem-focussed, designers are solution-focussed [18]. Due to these divergent approaches, the adaptation of LCA tools to design practices is still inadequate [17]. Designers are more interested in the relationship between the inputs and outputs, i.e., the association between the design decisions and the building performance, which needs improvement in the existing eco-design tools [16,19]. Additionally, the use of more sophisticated analysis methods, such as statistical uncertainty analysis, may miscommunicate the results to non-LCA experts [17]. Hence, designers prefer to rely on what they already know rather than performing simulations to explore other design possibilities [20]. There are a number of existing studies which focus on integrating LCA with 3D design model tools in the early stages of BPP [21–23], as well as a clear and graphical results visualisation [24]. However, unless it allows a coarser analysis of the main building systems or volumetry, connecting with 3D modelling is most valuable in the developed stages of BPP, when a greater amount of information is being generated [25]. In summary, understanding the software capabilities that are most valuable in a WBLCA should include designers' practice and perspective.

This research surveyed building designers who utilise Whole Building LCA (WBLCA) software simulations as a decision-making tool. The aim is to identify patterns in the software capabilities usage, thereby providing guidelines to WBLCA software developers. Ultimately, by improving designers' software experience, WBLCA will be integrated with the building project process (BPP) more efficiently.

Building LCA is an umbrella term that includes the LCA of individual construction materials, assemblies or building systems. WBLCA necessarily considers the building as a whole, including all parts and systems, as well as all stages of its life cycle. This approach is more effective, considering that buildings are a combination of multiple materials and processes that interact with each other. For instance, the selection of a specific structural system will narrow down the options of envelope assembly, which will affect the energy consumption during the use stage. The engineering systems in buildings determine the necessary construction processes, as well as the transportation distances from the suppliers. These are relevant factors in LCA, which should evolve to find better environmental solutions. The WBLCA approach is used since this research aims to investigate the systematic approach to design decisions.

Several surveys have been conducted to investigate and improve life cycle performance in the building design process from the designers' perspective [15,17,26]. Basbagill, Flager [26] conducted a charrette with designers to assess the efficacy of a range of software-generated graphical visualisations, demonstrating how much designers rely on these tools to make decisions. Bruce-Hyrkäs, Pasanen [27] conducted a survey to ascertain the challenges of adopting LCA in the building sector. It was found that opportunities to integrate software tools into the design process are vast, particularly when exploring BIM capabilities. In summary, a user's survey is an effective methodology to guide software tools to improve features and procedures most relevant to designers [28]. This is the first survey to investigate WBLCA software tools usage worldwide, exploring the most valuable software capabilities based on designers' location, professional background, years of WBLCA experience, assessment purpose and BPP stage.

This study is part of an ongoing PhD candidature. Previous publications have detailed the list of software attributes and informative outputs presented in the survey. The software attributes were based on a systematic literature review of peer-reviewed articles that applied a user-centred approach [25]. The informative outputs were gathered after analysing several WBLCA software tools whose target users were non-LCA experts [29]. This research reveals the current status of WBLCA practice, which will change over time as designers become experts, codes and legislation become updated and WBLCA matures. This is an important

step toward understanding the current demands, which informs the development of more design-oriented tools.

## 2. Materials and Methods

An online survey was undertaken between June and December of 2021, targeting building designers who use WBLCA software tools for decision-making. A variety of non-probability sampling methods were used, such as convenience, online "opt-in" through social media and snowball [30,31]. These sampling methods were selected because this research is exploring software usage on a global scale, developing an initial understanding of the population, which is an area with no pre-existing knowledge. For this purpose, the survey link was published on a professional networking and career development platform [32], where it was advertised in many groups related to the environmental impact of buildings. Potential users were searched in WBLCA software tools and Green Building Rating Systems (GBRS) webpages. They were contacted individually via the career development platform or e-mail. Non-respondents were contacted a second time with a follow-up message. The online survey was also published in the Carbon Leadership Forum community in its August 2021 newsletter [33]. The survey was not sponsored by any party and was conducted only by the authors of this manuscript. One hundred seventy-eight valid responses were collected.

As illustrated in Figure 1, the survey was structured in three parts: (1) demographic questions, (2) software attributes and (3) informative outputs. Part 1 gathered information about the participants' geographical location, professional background, duration of performing WBLCA, their motivation and at which stage of the Building Project Process (BPP) they usually performed the assessment.

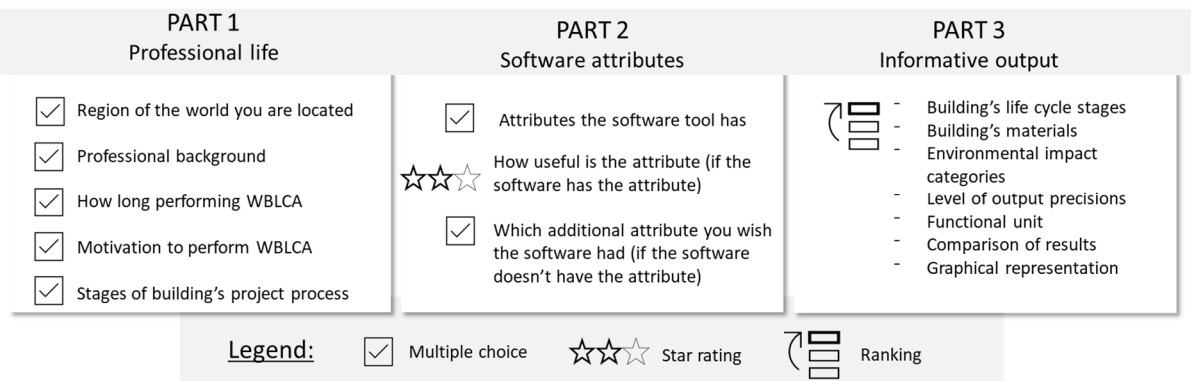

**Figure 1.** Survey flow.

In part 2, a list of 17 software attributes was presented. The Likert scale was used to rate software attributes in which 1 star represented "not useful at all" and 5 stars represented "extremely useful". Designers were able to nominate attributes that they wished were available (*WIH*). As indicated in Table 1, the attributes were categorised into 4 groups, design-oriented, inter-disciplinary connection, data-transparency and motivational. They were considered the most relevant attributes, according to an analysis of peer-reviewed articles that applied a user-centred approach [25].

In part 3, participants ranked in order of preference which output format would assist them in making more informed decisions. As shown in Table 2, the informative outputs were categorised as building life cycle stages, building materials, environmental impact categories, precision of outputs, functional unit, comparison of results and type of graphical representation. These informative outputs were based on a previous study on WBLCA software tools [34]. Graphical examples were provided so participants would have a better understanding of the alternatives. Participants took an average of 10 to 15 min to complete the survey, and the entire survey is displayed in Appendix A.

**Table 1.** List of attributes.

| Design-oriented | | Inter-disciplinary connection | |
|---|---|---|---|
| CODE | DESCRIPTION | CODE | DESCRIPTION |
| DO_1 | Compare results with other buildings or design alternatives | IC_1 | Comply with code or certification |
| DO_2 | Show graphical outputs | IC_2 | Provide an interface that adapts to the assessment scope |
| DO_3 | Provide a pre-set library with predefined WBLCA assumptions | IC_3 | Connect with a 3D model |
| DO_4 | Save previous design solutions | IC_4 | Combine LCA with cost analysis |
| DO_5 | Suggest design alternative | IC_5 | Connect with other building performance tools |

| Data-transparency | | Motivational | |
|---|---|---|---|
| CODE | DESCRIPTION | CODE | DESCRIPTION |
| DT_1 | Show numerical or table format outputs | M_1 | Provide technical support |
| DT_2 | Show the reasoning behind assumptions and results | M_2 | Provide fast or instant results |
| DT_3 | Allow you to edit the assumptions in a pre-set library | M_3 | Offer a free license version of the software |
| DT_4 | Indicates the range in which the results fall, instead of a single value | | |

**Table 2.** List of informative outputs.

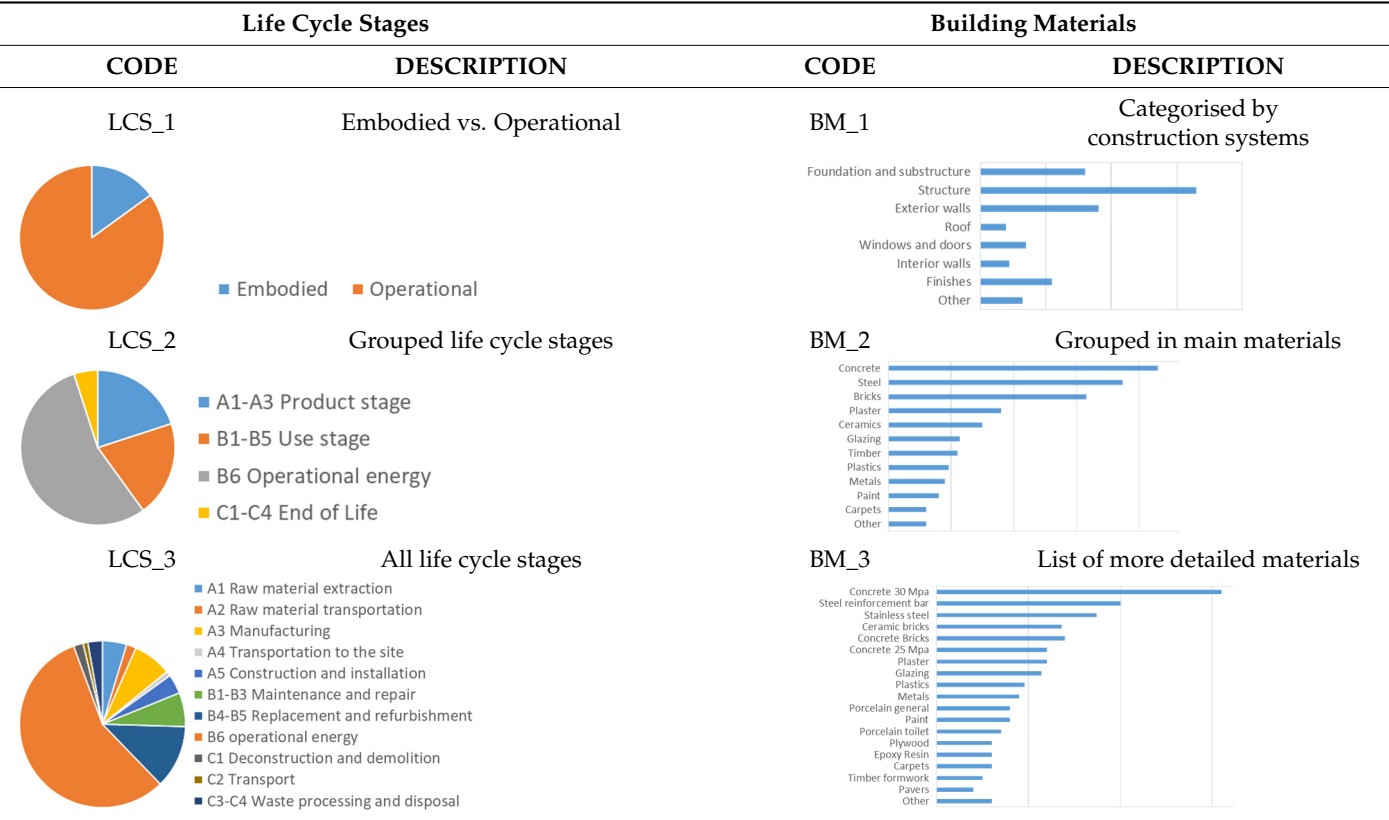

**Table 2.** *Cont.*

| Environmental impact categories | | Precision of outputs | |
|---|---|---|---|
| **CODE** | **DESCRIPTION** | **CODE** | **DESCRIPTION** |
| EIC_1 | Show only one impact category | PO_1 | Show the level of performance |
| EIC_2 | Show Endpoint impact categories  | PO_2 | Show the output range  |
| EIC_3 | Show Midpoint impact categories  | PO_3 | Show precise outputs  |

| Functional unit | | Comparison of results | |
|---|---|---|---|
| **CODE** | **DESCRIPTION** | **CODE** | **DESCRIPTION** |
| FU_1 | Whole building | CR_1 | I would prefer not to compare results |
| FU_2 | Per unit (e.g., per dwelling if residential or per office if commercial building) | CR_2 | Compare my design with a benchmark |
| FU_3 | Per metre square | CR_3 | Compare different design alternatives for the same building |
| FU_4 | Per occupant | CR_4 | Compare my design with other similar buildings |

| Graphical representation | | | |
|---|---|---|---|
| **CODE** | **DESCRIPTION** | **CODE** | **DESCRIPTION** |
| GR_1 | Pie | GR_2 | Bars |
| GR_3 | Columns | GR_4 | Radar |

Nine experts in WBLCA software development validated the quality of the survey, ensuring the eligibility of the terms used for the software capabilities. None of the questions were mandatory, i.e., participants were allowed to go to the next part of the survey without completely answering the previous one. The survey was considered for analysis only if all three parts of the survey had at least 50% of the questions answered.

Analysis of the data was performed in four steps. The first step analysed the relationship between the groups of participants presented in Table 3. The goal was to check if there was a significant relationship between them, e.g. if there is a difference in the years of experience between participants with architecture and engineering backgrounds.

**Table 3.** Groups of participants.

|  | Groups | Number of Participants (n) | % of Total (n = 178) |
|---|---|---|---|
| Regions of the world | Europe | 75 | 42.1% |
|  | North America | 51 | 28.7% |
|  | Australia | 21 | 11.8% |
|  | Rest of the world | 10 | 5.6% |
|  | Missing | 21 | 11.8% |
| Professional background | Architecture | 51 | 29.2% |
|  | Engineering | 75 | 42.1% |
|  | Missing/other | 51 | 28.7% |
| How long performing WBLCA | Less than 1 year | 32 | 18.0% |
|  | 1 to 5 years | 104 | 58.4% |
|  | 5 to 10 years | 25 | 14.0% |
|  | Over 10 years | 17 | 9.6% |
| Motivation to perform WBLCA | Client's request | 29 | 16.3% |
|  | Educations | 27 | 15.2% |
|  | Comply with certification | 56 | 31.5% |
|  | Personal reasons | 66 | 37.1% |
| Stages of BPP | Early stages (ES) | 74 | 41.6% |
|  | Developed stages (DS) | 53 | 29.8% |
|  | Handover stages (HS) | 28 | 15.7% |
|  | Use stage (US) | 11 | 6.2% |
|  | Missing/I do not know | 12 | 6.7% |

The second step analysed the software attributes participants wished the tool had. The wish it had (*WIH*) factor was then calculated using Equation (1). The higher the attribute's *WIH* factor, the more it is desired during the assessment. The *WIH* factor was compared within the groups of participants to check if there is a significant relationship between them, e.g. if there is a difference in the *WIH* factor of a certain attribute depending on the participants' motivations to perform WBLCA.

$$WIH = \frac{number\ of\ participants\ who\ wished\ the\ software\ tool\ had\ the\ attribute}{number\ of\ participants\ that\ claimed\ the\ software\ didn't\ have\ the\ attribute} \times 100. \quad (1)$$

The third step analysed the rating of the software attributes usefulness. The Kruskal–Wallis H test was used to identify the relationship between the ratings and the groups of participants. The Kruskal–Wallis H test only indicated that at least two groups were different. Therefore, when there were more than 2 groups, a post hoc Mann–Whitney test was used to compare all pairs of groups. The reason for choosing this method is the non-parametric nature of the Likert-scale questions. All the assumptions of a Kruskal–Wallis H test were met [35]:

1. The dependent variables (attributes' rating) are measured at the ordinal level;
2. The independent variable (groups of participants) consists of two or more categorical independent groups, as seen in Table 3;
3. The observations are independent, i.e., each group has different participants;
4. The distributions in each group have equal variability, i.e., they have negative skewness and similar standard deviation.

The last step of the data analysis looked at the ranking of the informative outputs and their relationship within the groups of participants. Although ranking questions also have a non-parametric nature, they did not meet Kruskal–Wallis assumption 4. Therefore, a chi-square test for independence was performed to check whether there was a difference in the informative output rankings within the groups of participants.

The statistical tests used in each of these steps are summarised in Table 4. A 95% confidence level (α = 0.05) was applied.

**Table 4.** Statistical tests performed in each step.

| Step | Analysis | Statistical Test | Level of Confidence ($\alpha$) |
|---|---|---|---|
| 1 | Relationship between the groups of participants | Chi-square test for independence | 0.05 |
| 2 | *WIH* factor of the software attributes | Chi-square test for independence | 0.05 |
| 3 | Usefulness ratings of the software attributes | Kruskal–Wallis H test | 0.05 |
| 4 | Ranking of the informative outputs | Chi-square test for independence | 0.05 |

## 3. Results and Discussion

The results follow the steps shown in Table 4, i.e., Section 3.1 shows the relationship between the groups of participants, Section 3.2 and 3.3 discuss the *WIH* factor and usefulness rating of the software attributes, respectively, and Section 3.4 discusses the ranking of the informative outputs.

### 3.1. Participants' Profile

Table 3 provide an overview of the participant's profile. In total, 42% were located in Europe and/or had an engineering background. Almost 60% of participants had been performing WBLCA for 1 to 5 years. The two most common reasons for performing WBLCA were complying with a certification, code or regulation (31.5%) and for personal reasons (37.1%). Among the personal reasons cited was the belief that it is the proper thing to do and the desire to gain a competitive advantage. Approximately 41.6% of participants performed WBLCA in the early stages of BPP, and as the project progressed to its final stages, the number of WBLCA practitioners reduced.

Table 5 show a cross-tabulation of the groups of participants only presenting significant results ($\alpha < 0.05$). In Australia and other countries outside of Europe and North America, there was a higher number of participants who had performed WBLCA for less than a year. This result suggests that WBLCA was more mature in European and North American countries. Performing WBLCA for certification or code compliance was more common in Europe than in other countries, where most participants performed WBLCA for personal reasons. European countries have been releasing codes that mandate life cycle analysis in the building industry, such as the Greater London Authority Whole Life Cabon [8], the Danish FBK (Den Frivillige Bæredygtighedsklasse (The Voluntary Sustainability Class)) [9], E+C-label [10] and RE 2020, the new French regulation for new buildings [11]. In addition to being largely from Europe, when WBLCA was motivated by code or certification compliance, most participants had an engineering background and had between 1 to 5 years of experience.

**Table 5.** Relationship between the groups of independent variables.

| | | How Long Performing WBLCA | | | | Motivation to Perform WBLCA | | | |
|---|---|---|---|---|---|---|---|---|---|
| | | Less than 1 Year | 1 to 5 Years | 5 to 10 Years | Over 10 Years | Client's Request | Educational | Comply with Certification | Personal Reasons |
| Regions of the world | Europe | 10.7% | 64.0% | 20.0% | 5.3% | 22.7% | 10.7% | 42.7% | 24.0% |
| | North America | 15.7% | 58.8% | 11.8% | 13.7% | 11.8% | 15.7% | 25.5% | 47.1% |
| | Australia | 33.3% | 47.6% | 0.0% | 19.0% | 19.0% | 14.3% | 9.5% | 57.1% |
| | Rest of the world | 40.0% | 50.0% | 0.0% | 10.0% | 20.0% | 30.0% | 10.0% | 40.0% |
| Professional background | Architecture | | | | | 9.6% | 23.1% | 23.1% | 44.2% |
| | Engineering | | | | | 28.0% | 6.7% | 37.3% | 28.0% |
| How long performing WBLCA | Less than 1 year | | | | | 12.5% | 3.1% | 31.3% | 53.1% |
| | 1 to 5 years | | | | | 11.5% | 17.3% | 41.3% | 29.8% |
| | 5 to 10 years | | | | | 36.0% | 20.0% | 12.0% | 32.0% |
| | Over 10 years | | | | | 23.5% | 17.6% | 0.0% | 58.8% |

The results show that while most participants with an architecture background performed WBLCA for educational and personal reasons, engineers performed it due to the client's request or to comply with code/certifications. Architects tended to focus on the building's aesthetics, functionality and volumetry [20], contrasting with the quantitative nature of WBLCA. Therefore, participants with an architecture background may have needed more time to become familiar with WBLCA before incorporating it into their practice. More than 50% of participants with less than 1 year of experience and those with more than 10 years of experience perform WBLCA for personal reasons.

### 3.2. Attributes' Wish It Had (WIH) Factor

The number of participants that responded whether the software tool has or does not have the desired attributes is shown in Figure 2. It also displays the number of participants that wished the software tool had the desired attributes; in case it did not.

The attributes software tools most have are within design-oriented and motivational categories, except for DO_5 (suggest design alternatives) and M_3 (Offer a free license version). One of the challenges of implementing DO_5 is maintaining a database with the design solutions typically used by designers, which varies depending on the type of building or geographical location. A shared database where designers can disclose the LCA results of their design solutions can facilitate the implementation of the DO_5 attribute [25]. In terms of offering a free license version of the software (M_3), there is a trade-off between this attribute and M_1 (provide technical support) due to the high cost of maintaining a dedicated user support team [25].

The attributes that are least available in software tools are within the inter-disciplinary connection and data-transparency groups, notably DT_4 (Indicates the range in which the results fall) and DT_2 (show the reasoning behind the results). Lack of these two attributes can potentially reduce the outputs' understanding and decrease designers' ability to make informed decisions. This is not desirable, especially if non-WBLCA specialists are within the software tools' target users.

The highest percentage of participants that wished the software tool had the attribute was within the inter-disciplinary connection category, notably IC_3 (connect with 3D model), IC_4 (combine LCA with cost analysis) and IC_5 (connect with other building performance tools). This shows that designers may desire connection between all aspects involved in the building's project, such as the 3D model, cost plan and other environmental impact assessment tools. DT_1 (show numerical or table format outputs), DO_4 (save previous design solutions) and M_3 (offer a free license version of the software) had the lowest *WIH* factor.

Table 6 show the *WIH* factors within the groups of participants. It only shows the attributes with statistically significant results ($\alpha < 0.05$), i.e., results where the *WIH* were significantly different among the groups. European countries showed a higher *WIH* factor in IC_4 (combine LCA with cost analysis) ($\chi^2(3, 92) = 8.3$, $p = 0.041$), while North American countries showed a higher *WIH* factor for DT_1 (show numerical or table format output) ($\chi^2(3, 43) = 9.3$, $p = 0.025$). DO_2 (show graphical outputs) had a greater *WIH* factor among architects ($\chi^2(1, 33) = 6.8$, $p = 0.009$). This result was expected since graphical outputs are the most effective way of making the results intelligible to designers, especially the ones with an architecture background [36]. Participants with less than 1 year of WBLCA experience had a higher *WIH* factor in M_3 (offer a free license of the software) ($\chi^2(3, 121) = 11.5$, $p = 0.009$). This result suggests that participants with fewer years of experience are still becoming familiar with the WBLCA methodology, and offering a free license of the software might encourage them to experiment with different tools available in the market to find the most compatible with their practice.

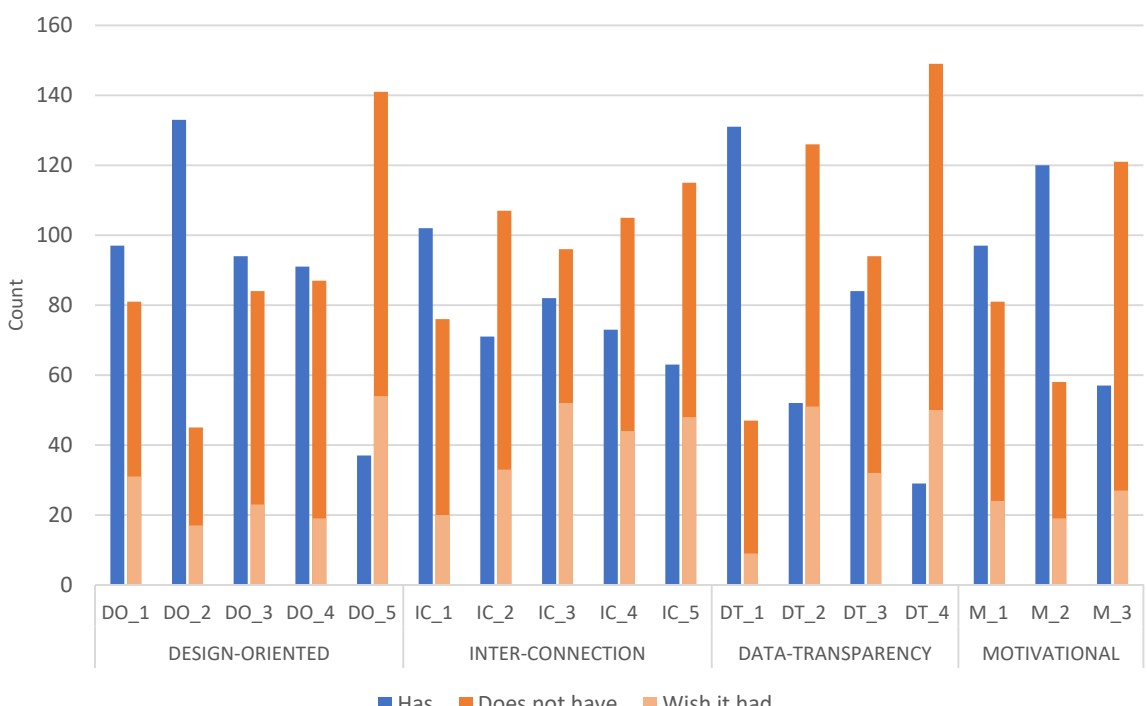

| | | Design-oriented | | | | | Inter-disciplinary connection | | | | | Data-transparency | | | | Motivational | | |
|---|---|---|---|---|---|---|---|---|---|---|---|---|---|---|---|---|---|---|---|
| | | DO_1 | DO_2 | DO_3 | DO_4 | DO_5 | IC_1 | IC_2 | IC_3 | IC_4 | IC_5 | DT_1 | DT_2 | DT_3 | DT_4 | M_1 | M_2 | M_3 |
| ■ | Has | 97 | 133 | 94 | 91 | 37 | 102 | 71 | 82 | 73 | 63 | 131 | 52 | 84 | 29 | 97 | 120 | 57 |
| ■ | Do not have | 81 | 45 | 84 | 87 | 141 | 76 | 107 | 96 | 105 | 115 | 47 | 126 | 94 | 149 | 81 | 58 | 121 |
| ■ | Wish it had (*WIH*) | 31 | 17 | 23 | 19 | 54 | 20 | 33 | 52 | 44 | 48 | 9 | 51 | 32 | 50 | 24 | 19 | 27 |
| | *WIH* factor (%) | 38.3 | 37.8 | 27.4 | 21.8 | 38.3 | 26.3 | 30.8 | 54.2 | 41.9 | 41.7 | 19.1 | 40.5 | 34.0 | 33.6 | 29.6 | 32.8 | 22.3 |

**Design-oriented:** DO_1 (Compare results with other buildings or design alternatives); DO-2 (Show graphical outputs); DO_3 (Provide a pre-set library with predefined WBLCA assumptions); DO_4 (Save previous design solutions); DO_5 (Suggest design alternative); **Inter-disciplinary connection:** IC_1 (Comply with code or certification); IC_2 (Provide an interface that adapts to the assessment scope); IC_3 (Connect with 3D model); IC_4 (Combine LCA with cost analysis); IC_5 (Connect with other building performance tools); **Data-transparency:** DT_1 (Show numerical or table format outputs); DT_2 (Show the reasoning behind assumptions and results); DT_3 (Allow you to edit the assumptions in a pre-set library); DT_4 (Indicates the range in which the results fall, instead of a single value); **Motivational:** M_1 (Provide technical support); M_2 (Provide fast or instant results); M_3 (Offer a free license version of the software)

**Figure 2.** Total of participants that assigned whether the software tool has, does not have or wish it had the attribute.

**Table 6.** Wish it had (*WIH*) percentages within different groups.

| Independent Variables | Groups | Design-Orient. | Inter-Disciplinary Connection | | | Data-Transparency | | Motivational | |
|---|---|---|---|---|---|---|---|---|---|
| | | DO_2 | IC_1 | IC_4 | IC_5 | DT_1 | DT_2 | M_2 | M_3 |
| | | Show Graphical Outputs | Comply with Code or Certification | Combine LCA with Cost Analysis | Connect with Other Building Performance Tools | Show Numerical or Table Forma Outputs | Show the Reasoning behind Assumptions and Results | Provide Fast or Instant Results | Offer a Free License Version of the Software |
| Regions of the world | Europe | | | 54.8% | | 6.3% | | | |
| | North America | | | 33.3% | | 25.0% | | | |
| | Australia | | | 10.0% | | 18.2% | | | |
| | Rest of the world | | | * | | * | | | |
| Professional background | Architecture | 68.8% | | | | | | | |
| | Engineering | 23.5% | | | | | | | |
| How long performing WBLCA | Less than 1 year | | | | | | | | 46.2% |
| | 1 to 5 years | | | | | | | | 15.5% |
| | 5 to 10 years | | | | | | | | 23.1% |
| | Over 10 years | | | | | | | | 9.1% |
| Motivation to perform WBLCA | Client's request | | | | 20.0% | | 26.3% | 15.4% | |
| | Educational | | | | 29.4% | | 50.0% | 18.2% | |
| | Comply with certification | | | | 55.6% | | 28.6% | 64.3% | |
| | Personal reasons | | | | 45.2% | | 53.7% | 30.0% | |
| Stages of BPP | Early Stage | | | | | 5.3% | | | |
| | Developed Stage | | | | | 35.7% | | | |
| | Handover Stage | | | | | 50.0% | | | |
| | Use Stage | | | | | * | | | |

* Number of cases below 5.

Participants that performed WBLCA for educational and personal purposes had a higher *WIH* factor in DT_2 (show the reasoning behind assumptions and results) ($\chi^2(3, 126) = 7.9$, $p = 0.048$). This suggests that both groups were willing to investigate the calculation process in more detail. Although this is a time-consuming task, understanding the reasoning behind the results would allow participants to replicate the assessment with different tools. Those who perform WBLCA for code/certification reasons indicated a higher *WIH* factor in IC_5 (connect with other building performance tools) ($\chi^2(3, 115) = 8.0$, $p = 0.046$) and M_2 (provide fast or instant results) ($\chi^2(3, 58) = 9.2$, $p = 0.026$). Both attributes may facilitate code/certification compliance by providing timely reports.

Show numerical or table format outputs (DT_1) applied to participants who performed WBLCA during the handover stage of BPP ($\chi^2(3, 41) = 8.0$, $p = 0.047$). In the handover stages, contractors may suggest cost-effective materials with similar environmental profiles [25]. Therefore, numerical or table format outputs would highlight the changes available without compromising the sustainability targets established in previous stages of BPP [2].

*3.3. Rating the Usefulness of Attributes*

Figure 3 show the rating for each one of the attributes, as well as the average for the ratings. Overall, the averages lie between 3.4 and 4.4, and the high number of 4 and 5 stars indicate that the provided set of attributes was indeed useful for WBLCA assessment.

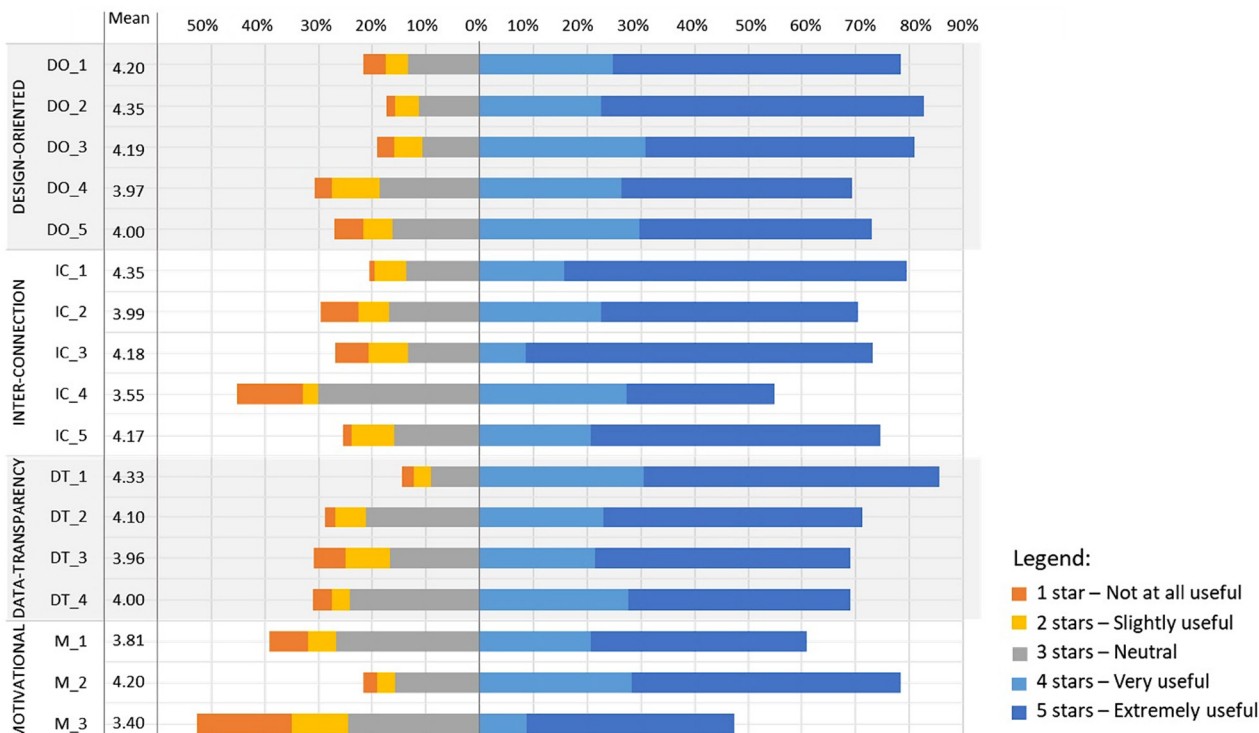

**Figure 3.** Attribute's star rating. **Design-oriented:** DO_1 (Compare results with other buildings or design alternatives); DO-2 (Show graphical outputs); DO_3 (Provide a pre-set library with predefined WBLCA assumptions); DO_4 (Save previous design solutions); DO_5 (Suggest design alternative); **Inter-disciplinary connection:** IC_1 (Comply with code or certification); IC_2 (Provide an interface that adapts to the assessment scope); IC_3 (Connect with 3D model); IC_4 (Combine LCA with cost analysis); IC_5 (Connect with other building performance tools); **Data-transparency:** DT_1 (Show numerical or table format outputs); DT_2 (Show the reasoning behind assumptions and results); DT_3 (Allow you to edit the assumptions in a pre-set library); DT_4 (Indicates the range in which the results fall, instead of a single value); **Motivational:** M_1 (Provide technical support); M_2 (Provide fast or instant results); M_3 (Offer a free license version of the software).

M_3 (offer a free license version of the software) and IC_4 (combine LCA with cost analysis) are considered the least useful. Although IC_4 had a lower average usefulness rate, it does not align with the results from Section 3.2, where combining LCA with cost analysis had a high *WIH* factor. This suggests that, although combining cost with LCA is highly desirable, developers might be struggling to implement IC_4 in a way that suits their current users.

More than 80% of participants agreed that DO_2 (show graphical outputs) and DT_1 (show numerical or table format outputs) are very useful and extremely useful. These attributes could support user understanding of the results. DO_3 (provide a pre-set library with predefined WBLCA assumptions) also falls in this group, potentially improving the efficiency of information entry. More than half of the participants agreed that IC_3 (connect with 3D modelling) and IC_5 (connect with other building performance tools) are extremely useful. These two attributes could also increase the efficiency of usage. Figure 2 indicate that these are the attributes with high *WIH* factors. However, they were missing from most participants' software tools. This result indicates that software tools should focus on the development of inter-disciplinary connection attributes, especially IC_3 and IC_5.

Table 7 show the attribute's mean within the groups where the p-value was lower than 0.05 ($\alpha < 0.05$).

**Table 7.** Rating means of the software attributes' usefulness within different groups.

| Independent Variable | Groups | Design-Oriented | | Inter-Connection | | | Data-Transparency | | Motivational | | |
| | | DO_1 | DO_2 | IC_1 | IC_4 | IC_5 | DT_3 | DT_4 | M_1 | M_2 | M_3 |
| | | Compare Results with Other Buildings or Design Alternatives | Show Graphical Outputs | Comply with Code or Certification | Combine LCA with Cost Analysis | Connect with Other Building Performance Tools | Allow you to Edit the Assumptions in a Pre-Set Library | Indicates the Range in Which the Results Fall, instead of a Single Value | Provide Technical Support | Provide Fast or Instant Results | Offer a Free License Version of the Software |
|---|---|---|---|---|---|---|---|---|---|---|---|
| Regions of the world | Europe | | | | | 4.48 | | | | | |
| | North America | | | | | 3.63 | | | | | |
| | Australia/ New Zealand | | | | | * | | | | | |
| | Rest of the world | | | | | * | | | | | |
| Professional background | Architecture | | | | | | | 3.40 | | | 3.95 |
| | Engineering | | | | | | | 4.35 | | | 2.95 |
| How long performing WBLCA | <1 year | | | 3.71 | 4.12 | | 3.82 | | 4.22 | | |
| | 1 to 5 years | | | 4.44 | 3.84 | | 3.96 | | 3.94 | | |
| | 5 to 10 years | | | 4.79 | 2.30 | | 4.89 | | 4.06 | | |
| | >10 years | | | 4.29 | 2.67 | | 3.25 | | 2.18 | | |
| Motivation to perform WBLCA | Client's request | | | | | | | | | | 2.75 |
| | Educational | | | | | | | | | | 4.45 |
| | Comply with certification | | | | | | | | | | 2.95 |
| | Personal reasons | | | | | | | | | | 3.56 |
| Stages of BPP | Early (ES) | 4.47 | | | 4.03 | | | | | 4.31 | |
| | Developed (DS) | 4.17 | | | 4.68 | | | | | 4.21 | |
| | Handover (HS) | 3.27 | | | 4.65 | | | | | 4.00 | |
| | Use (US) | 4.00 | | | 3.20 | | | | | 3.14 | |

* Number of cases below 5.

European countries showed higher usefulness mean rank for IC_5 (connect with other building performance tools) compared with North American countries (U = 11.79, $p = 0.004$). European countries may find this attribute useful for code/certification compliance since this is the main motivation to perform WBLCA in Europe (Table 5), and connecting with other building performance tools is the most desirable attribute for those who perform WBLCA for code/certification purposes (Table 6).

Participants with an architecture background (U = 5.1, $p = 0.024$) and whose motivation is educational or personal reasons (H(3) = 10.0, $p = 0.019$) find M_3 (offer a free license version of the software) more useful than the other groups. As revealed in Table 5, participants with an architectural background and with a lower level of understanding perform WBLCA mostly for personal reasons. There may not be any commercial return for these users, so a free license version of the software would encourage these groups to apply WBLCA in their projects, even if the functionality was limited in some way. Participants with an engineering background find DT_4 (Indicate the range in which the results fall, instead of a single value) more useful than architects (U = 4.2, $p = 0.041$). This may be due to the quantitative nature of engineering, which is more familiar with this type of results representation.

Participants that have been performing WBLCA between 5 to 10 years consider DT_3 (allowing you to edit the assumptions in a pre-set library) more useful than other groups with less or more years of experience (H(3) = 8.03, *p* = 0.045). Editing the assumptions requires a higher level of WBLCA understanding since the user can modify the calculation variables. The modifications would also need to be included in any generated reports.

The results suggested that the most useful attributes in each stage of BPP are related to the activities performed in each stage. In the early stages, when design possibilities are being investigated through feasibility analysis, DO_1 (compare results with other buildings or design alternatives) becomes an effective attribute (H(3) = 10.4, *p* = 0.016). While design solutions are being validated, tested and detailed in the early and developed stages, communication among the project's stakeholders becomes a key aspect [37]. Fast or instant results (M_2) can potentially make this communication more effective (H(3) = 8.2, *p* = 0.042). Although code or certification compliance starts right at the beginning of the BPP, documentation is mostly prepared during the developed and handover stages [25]. Therefore, IC_1 (Comply with code or certification) would become more useful during those stages (H(3) = 12.6, *p* = 0.006) to produce and submit reports according to the code/certification requirements.

### 3.4. Output Preferences

Figure 4 show the percentages of the rankings for each one of the output preferences, and Table 8 show the proportion of the most preferred output within the groups of participants.

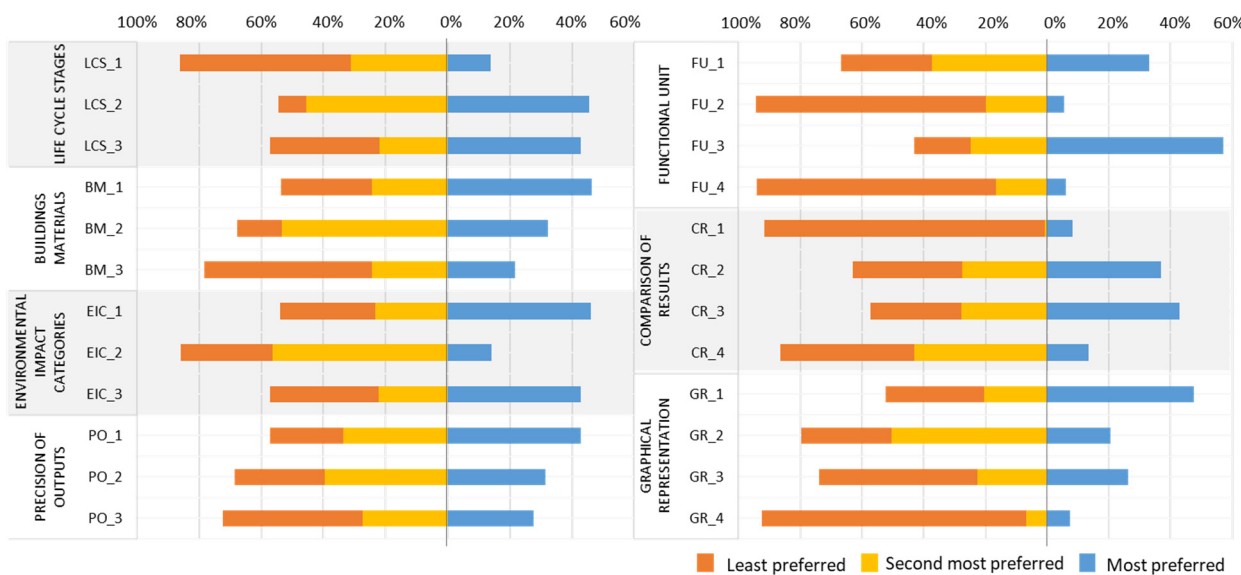

**Figure 4.** Ratings for the output's preferences for decision-making. **Life cycle stages:** LCS_1 (Embodied vs. Operational); LCS_2 (Grouped life cycle stages); LCS_3 (All life cycle stages). **Building materials:** BM_1 (Categorizsd by construction systems); BM_2 (Grouped in main materials); BM_3 (List of more detailed materials). **Environmental impact categories:** EIC_1 (Show only one impact category); EIC_2 (Show Endpoint impact categories); EIC_3 (Show Midpoint impact categories). **Precision of outputs:** PO_1 (Show the level of performance); PO_2 (Show the output range); PO_3 (Show precise outputs). **Functional unit:** FU_1 (Whole building); FU_2 (Per unit); FU_3 (Per metre square); FU_4 (Per occupant). **Comparison of results:** CR_1 (I would prefer not to compare results); CR_2 (Compare my design with a benchmark); CR_3 (Compare different design alternatives of the same building); CR_4 (Compare my design with other similar buildings). **Graphical representation:** GR_1 (Pie); GR_2 (Columns); GR_3 (Bars); GR_4 (Radar).



**Table 8.** Percentages of preferred outputs within different groups of independent variables.

| | Groups | Life Cycle Stages | | | Buildings Materials | | Envir. Impact Cate-gories | Precision of Outputs | | Functional Unit | | Comparison of Results | | Graphical Represent. | |
|---|---|---|---|---|---|---|---|---|---|---|---|---|---|---|---|
| | | LCS_1 | LCS_2 | LCS_3 | BM_1 | BM_2 | EIC_3 | PO_2 | PO_3 | FU_1 | FU_3 | CR_2 | CR_3 | GR_2 | GR_3 |
| | | Embodied vs. Operational | Grouped Life Cycle Stages | All Life Cycle Stages | Categorised by Construction Systems | Grouped in Main Materials | Show Midpoint Impact Categories | Show the Output Range | Show Precise Outputs | Whole Building | Per Metre Square | Compare my Design with a Benchmark | Compare Different Design Alternatives of the same building | Bars | Columns |
| Regions of the world | Europe | 4.5% | | | 56.5% | | 48.6% | | | | | 27.9% | | | |
| | North America | 18.2% | | | 31.8% | | 31.0% | | | | | 45.5% | | | |
| | Australia | 25.0% | | | 31.6% | | 56.3% | | | | | 35.0% | | | |
| | Rest of the world | 25.0% | | | 25.0% | | 10.0% | | | | | 77.8% | | | |
| Professional background | Architecture | | | | 53.3% | 26.1% | | | | 40.9% | 48.9% | | | 33.3% | 10.9% |
| | Engineering | | | | 31.3% | 44.6% | | | | 18.8% | 67.7% | | | 15.4% | 37.5% |
| How long performing WBLCA | Less than 1 year | | 65.5% | 20.0% | | | | | | | | 53.3% | | | |
| | 1 to 5 years | | 51.1% | 38.0% | | | | | | | | 37.0% | | | |
| | 5 to 10 years | | 21.7% | 69.6% | | | | | | | | 43.5% | | | |
| | Over 10 years | | 14.3% | 62.5% | | | | | | | | 0.0% | | | |
| Motivation to perform WBLCA | Client's request | | | | | | | 32.0% | 30.8% | 16.0% | 84.6% | 56.0% | | | |
| | Educational | | | | | | | 50.0% | 25.0% | 76.5% | 23.8% | 16.0% | | | |
| | Comply with certification | | | | | | | 19.6% | 39.2% | 30.0% | 65.3% | 41.2% | | | |
| | Personal reasons | | | | | | | 37.9% | 13.6% | 30.2% | 49.1% | 35.6% | | | |
| Stages of BPP | Early Stage | | | | | | | | | | | | 53.7% | | |
| | Developed Stage | | | | | | | | | | | | 45.2% | | |
| | Handover Stage | | | | | | | | | | | | 14.8% | | |
| | Use Stage | | | | | | | | | | | | 36.4% | | |

Overall, most participants would prefer to see grouped life cycle stages (LC_2), followed by a more detailed list with all life cycle stages (LC_3). Most Green Building Rating Systems (GBRS) only mention the system boundary, not specifying how detailed the various life cycle stages need to be. Some tools leave this decision up to the users, providing flexibility on how they would like to see the total impacts in each life cycle stage [29]. Although LCS_1 (embodied vs. operational) was not the most preferred method overall, countries outside of Europe were more likely to prefer this type of output ($\chi^2(3, 138) = 8.8$, $p = 0.033$). Results suggest that the level of detail of the life cycle stages is proportional to how long participants have been performing WBLCA. Higher years of experience prefer to see the impact in all life cycle stages (LCS_3) ($\chi^2(3, 161) = 16.5$, $p = 0.001$), and lower years of experience prefer the impacts grouped in main life cycle stages (LCS_2) ($\chi^2(3, 160) = 16.5$, $p = 0.001$).

Regarding building materials, almost 50% of participants would prefer to categorise them by construction systems (BM_1). This type of output is accepted in most GBRS [34]. Participants from European countries ($\chi^2(3, 140) = 9.4$, $p = 0.024$) and with an architecture background ($\chi^2(1, 112) = 5.4$, $p = 0.02$) preferred this output method. However, 45% of participants with an engineering background preferred to see the impacts grouped in main materials (BM_2) ($\chi^2(1, 140) = 4.0$, $p = 0.046$). The more detailed the list of materials, the greater influence the user has over the variables, as it reveals which material is impacting the most.

Regarding the environmental impact categories (EIC), most participants would prefer the output with only one impact category (EIC_1). Global Warming Potential (GWP) is the most commonly used EIC [38,39], and some Green Building Rating Systems (GBRS) only require the assessment of GWP [40,41]. The standards that outline the calculation for the assessment of the environmental performance of buildings [12] and the elaboration of the Environmental Product Declaration (EPD) [42] specify the application of midpoint impact categories (EIC_3), which was the preferred method for most participants located in Europe and Australia ($\chi^2(3, 138) = 8.9$, $p = 0.031$). This result suggests that in terms of environmental impact categories, Europe and Australia are more closely aligned with the international standards.

Regarding the precision of outputs, overall, participants preferred the outputs showing the level of performance (PO_1), which is the most imprecise method. This suggests that participants are not interested in the exact number but in which impact range the building is located compared to the relevant scale. When WBLCA is performed for educational purposes, half of the participants from this group preferred that the tool show the output range (PO_2) ($\chi^2(3, 160) = 8.2$, $p = 0.043$). The output range communicates the uncertainties inherent in the assessment, refining the comparative analysis among the design options [43]. Participants that perform WBLCA for code/certification purposes are more likely to prefer precise outputs (PO_3) ($\chi^2(3, 160) = 9.6$, $p = 0.022$). In fact, most GBRS does not require the treatment of the uncertainties, as proposed by the ILCD handbook [44]. The results suggest that GBRS are not conforming in their treatment and communication of the uncertainties in the assessment, potentially leading to misinterpretation of the results [45].

In terms of functional unit, 57% of participants would prefer to see the impacts per metre square (FU_3), especially the respondents with an engineering background ($\chi^2(1, 112) = 4.0$, $p = 0.046$) and who perform WBLCA for client's request or code/certification compliance ($\chi^2(3, 149) = 20.3$, $p < 0.001$). Table 5 show that these groups are related, i.e., most engineers perform WBLCA for the above reasons. Results per metre square allow users to compare their design with similar buildings or benchmarks, which is a method mostly required by GBRS [46–49]. Participants that perform WLCA for educational reasons would prefer the whole building as the functional unit (FU_1) ($\chi^2(3, 145) = 18.2$, $p < 0.001$). Although it is harder to compare WBLCA results of different buildings' typologies, the functional unit whole building can be more versatile. While the whole building can be easily converted into a metre square, the opposite is not always true since it may be unclear whether the area is of conditioned area, total or Net Lettable Area (NLA), for example.

Regarding the comparison of results, overall, 43% of participants would prefer to compare different design alternatives of the same building (CR_3), particularly during the early stages of the building project process (BPP) ($\chi^2$(3, 147) = 12.2, $p$ = 0.007), when this strategy is used to select the best design options. Comparing the design with a benchmark (CR_2) was the most preferred for participants with less than 10 years of experience ($\chi^2$(3, 160) = 12.6, $p$ = 0.006) and for those who perform WBLCA at the client's request and for code/certification compliance ($\chi^2$(3, 160) = 9.0, $p$ = 0.03). Benchmarks play a relevant role in the decision-making process [37,50,51]. Many sectors of the construction industry are focused on its development, such as software tools [29], certification systems [7] and country-specific industry benchmarks [50–52]. Global initiatives are used as resources for key performance indicators, such as the UN 2030 Sustainable Development Goals [53] and the Paris Agreement [54]. However, the development of regional benchmarks would provide a more accurate means of comparison [55].

In terms of graphical representation, 48% of participants would prefer the result in a pie chart (GR_1). Looking at the groups of participants, those with an architecture background are more likely to prefer bar charts (GR_2) ($\chi^2$(1, 113) = 5.0, $p$ = 0.025), while those with an engineering background are more likely to prefer columns (GR_3) ($\chi^2$(1, 110) = 9.8, $p$ = 0.002).

## 4. Conclusions

This paper analysed a survey of building designers, e.g., engineers and architects, who utilise WBLCA software simulations as a decision-making tool. The aim was to identify patterns in the software capabilities usage to provide guidelines to WBLCA software developers. Ultimately, by improving the software experience of designers, WBLCA will be integrated with BPP more efficiently. The survey drew the user's profiles identifying their region of the world, professional backgrounds, years of WBLCA experience, assessment purposes and at which stage of BPP they perform the assessment. Participants rated the software attributes they found useful and selected the ones they would like to see in the software tools. Different categories of informative outputs were listed, and participants ranked their preferences.

The most valuable software attributes and informative outputs found within the groups of participants were summarised into a relationship map, as indicated in Figure 5 and summarised in Table 9. Two main groups of participants were identified. Each group has its software capability preferences. Knowing users' preferences will point software developers in the right direction, indicating where they should concentrate their effort.

The first group is centred around performing WBLCA for code or certification compliance. Most of the participants that perform WBLCA for code or certification compliance are from European countries and have between 1 to 5 years of experience and engineering background. In addition to code or certification compliance, most engineers also perform WBLCA at their clients' request. The second group is centred around performing WBLCA for personal reasons, which includes the belief that it is the proper thing to do and the desire to gain a competitive advantage. Most participants that perform WBLCA for personal reasons are from North America or Australia and have less than 1 year or over 10 years of experience and architectural background. Most architects, besides performing WBLCA for personal reasons, also do it for educational purposes. Most participants with less than 1 year of experience are from countries outside North America and Europe. This result shows that European countries are further ahead in terms of developing codes that mandate LCA before issuing building permits. The formal education of architects in many countries is primarily centred on fine arts, leaving the quantitative role to engineers, which is the ability most required in a WBLCA. Therefore, architects may need more time to educate themselves before incorporating WBLCA into their practice.

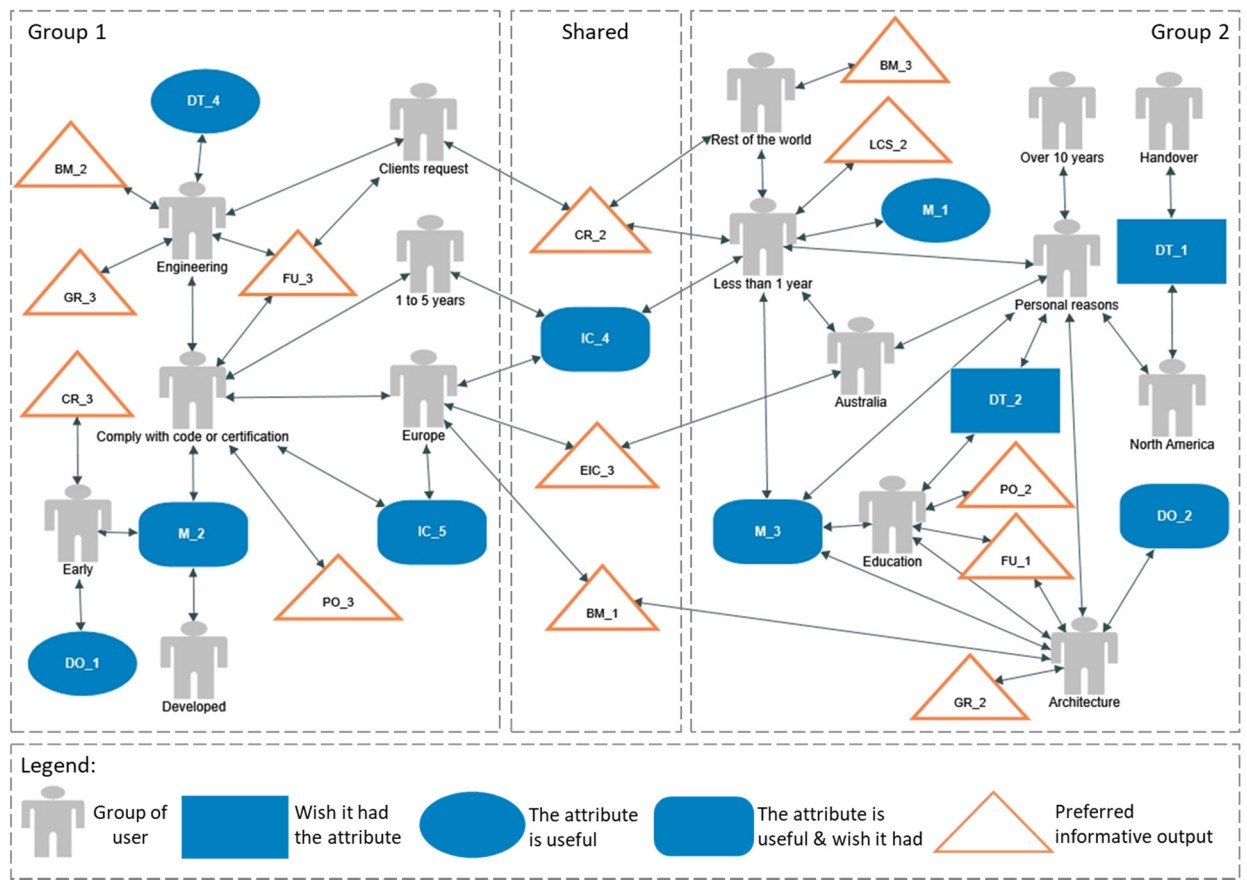

**Figure 5.** Relationship map. **ATTRIBUTES: Design-oriented:** DO_1 (Compare results with other buildings or design alternatives); DO-2 (Show graphical outputs); **Inter-disciplinary connection:** IC_4 (Combine LCA with cost analysis); IC_5 (Connect with other building performance tools); **Data-transparency:** DT_1 (Show numerical or table format outputs); DT_2 (Show the reasoning behind assumptions and results); DT_4 (Indicates the range in which the results fall, instead of a single value); **Motivational:** M_1 (Provide technical support); M_2 (Provide fast or instant results); M_3 (Offer a free license version of the software). **OUTPUT: Life cycle stages:** LCS_2 (Grouped life cycle stages); **Building materials:** BM_1 (Categorised by construction systems); BM_2 (Grouped in main materials); BM_3 (List of more detailed materials). **Environmental impact categories:** EIC_3 (Show Midpoint impact categories). **Precision of outputs:** PO_2 (Show the output range); PO_3 (Show precise outputs). **Functional unit:** FU_1 (Whole building); FU_3 (Per metre square). **Comparison of results:** CR_2 (Compare my design with a benchmark); CR_3 (Compare different design alternatives of the same building). **Graphical representation:** GR_2 (Columns); GR_3 (Bars).

**Table 9.** Guidelines based on the groups of participants.

| Software Attributes | Informative Output | Group of Participants | Guidelines |
|---|---|---|---|
| | | GROUP 1 | |
| IC_5 (connect with other building performance tool) | | Europe<br>Comply with code or certification | The connection with a building energy performance tool should be developed for code compliance, especially for designers in European countries. |
| M_2 (provide fast or instant results) | | Comply with code or certification<br>Early stages<br>Developed stages | Provide updated and timely reports for both code and certification compliance. This is also relevant in the stages of BPP when design changes are constantly being made to achieve the environmental performance target. |

**Table 9.** *Cont.*

| Software Attributes | Informative Output | Group of Participants | Guidelines |
|---|---|---|---|
| DO_1 (Compare results with other buildings or design alternatives) | CR_3 (Compare different design alternatives of the same building). | Early stages | Allow users to compare different design alternatives in the early stages of BPP when there is the highest level of uncertainty compared to other stages. The comparison should provide an overview of where the building stands on the environmental impact scale instead of a precise number. |
| | PO_3 (Show precise outputs). | Comply with code or certification | Show precise outputs when users perform WBLCA for code or certification purposes. |
| | FU_3 (Functional unit per metre square) | Comply with code or certification Client's request Engineering background | When complying with a code or certification, results should be shown in square metres. This will allow users to compare their design with similar buildings or benchmarks, which is a method mostly required by GBRS. |
| DT_4 (Indicates the range in which the results fall, instead of a single value); | GR_3 (Bars), BM_2 (Building materials grouped in main materials); | Engineering background | Engineers should be provided with the output range, communicating the uncertainties inherent in the assessment and refining the comparative analysis among the design options. |
| GROUP 2 | | | |
| M_3 (Offer a free license version of the software). | | Education Personal reasons Architecture background Less than 1 year of experience | Offer a free license of the software to designers who are still becoming familiar with the WBLCA methodology. This will encourage them to experiment with different tools available on the market to find the most compatible with their practice. |
| DT_1 (Show numerical or table format outputs); | | North America Handover | Numerical or table format outputs should be available when WBLCA is performed during the handover stages. This will help contractors to suggest cost-effective materials without compromising the sustainability targets established in previous stages of BPP. This attribute is also more desirable when WBLCA is mainly focused on embodied emissions, such as in North American countries. |
| DT_2 (Show the reasoning behind assumptions and results); | | Personal reasons Education | For those who utilise the software for educational purposes, the software should show the reasoning behind the assumptions, allowing users to understand and replicate the results. Transparency is desired in research projects when the methodology adopted should be clearly disclosed. |
| | FU_1 (Whole building) | Architecture background Education | This group of designers prefer the whole building as a functional unit (FU). This will allow them to compare their results based on the building's typology, such as residential or commercial. Another factor is the versatility that this FU provides. While the whole building can be easily converted into $m^2$, the opposite is not always true since it may be unclear whether the $m^2$ is of conditioned area, total or Net Lettable Area (NLA), for example. |

**Table 9.** *Cont.*

| Software Attributes | Informative Output | Group of Participants | Guidelines |
|---|---|---|---|
| DO_2 (Show graphical outputs) | GR_2 (Columns) | Architecture background | Architects should see the results in a graphical manner. |
| | PO_2 (Show the output range) | Education | Those who perform WBLCA for educational purpose should be informed about the results' range. In fact, a common research topic is predicting the uncertainties of an LCA [56–58]. |
| M_1 (Provide technical support) | LCS_2 (Life cycle stages clustered in main groups) | Less than 1 year | Technical support and a simplification of outputs are required for those with less than 1 year of WBLCA. |
| SHARED | | | |
| | CR_2 (Compare my design with a benchmark) | Clients' request<br>Less than 1 year of experience<br>Rest of the world | Benchmarks play a relevant role in the decision-making process. Regional benchmarks should be developed to provide a more accurate means of comparison |
| IC_4 (Combine LCA with cost analysis) | | Less than 1 year of experience<br>1 to 5 years of experience<br>Europe | Combining LCA with the cost is extremely relevant for both groups of software users identified. However, developers are still struggling to properly implement IC_4, as users who experience this attribute find it less useful. |
| | EIC_3 (Show Midpoint impact categories). | Australia<br>Europe | LCA standards specify the application of midpoint impact categories |
| | BM_1 (Building materials categorised by construction systems) | Architecture background<br>Europe | This type of output is accepted in most GBRS [34]. It is also a simplified way of showing the results of the building materials' impacts. |

## 5. Limitations and Future Developments

The survey developed targeted designers who use WBLCA software tools to make decisions, i.e., non-LCA specialists. Similarly, a survey could investigate the software tools that LCA experts use to find out their specific requirements.

This research acknowledges the unmeasured errors associated with all forms of public opinion research [31], such as those related to the selection of respondents and the accuracy of the measured responses [59]. For instance, it is unknown why participants have not answered specific questions. Given that the majority of participants are from countries where English is not the native tongue, some questions might have been misunderstood.

The guidelines provided are aligned with the current design practices. Design activities are constantly changing to adapt to the building's codes, regulations and certification requirements. Therefore, this research acknowledges the impermanence character of the results. This exploratory research took an innovative step toward understanding WBLCA practices on a global scale, leading to the development of more design-oriented tools. Similar surveys can be performed on a smaller scale, considering the influence of country-specific LCA requirements on decision-makers' practices.

**Author Contributions:** T.S.: Conceptualization, Methodology, Writing, Data Analysis, Submission Management; R.D.: Conceptualization, Review and Editing, Supervision, Refinement of conclusions; S.O.: Conceptualization, Review and Editing, Supervision, Refinement of Conclusions. F.L.: Review and Editing, Supervision. All authors have read and agreed to the published version of the manuscript.

**Funding:** This research received no external funding.

**Institutional Review Board Statement:** This research was authorised by the ethical committee of the Queensland University of Technology (QUT) with approval number 2000000956.

**Informed Consent Statement:** Informed consent was obtained from all subjects involved in the study.

**Data Availability Statement:** The data presented in this study are available on request from the corresponding author. The data are not publicly available due to ethical restrictions.

**Acknowledgments:** This research was supported by Building 4.0 CRC (Cooperative Research Centre).

**Conflicts of Interest:** The authors declare no conflict of interest.

**Appendix A**

Online survey
Q2 Select the region of the world you are located.
▼ Africa (1) . . . South America (7)
Q3 What is your professional background?

○ Architecture (1)
○ Urban designer (2)
○ Engineering (3)
○ Project Management (4)
○ Surveyor (5)
○ I'm a student. Please specify your major: (6) ________________________________
○ Other: (7) ________________________________________

Q4 Which of the below best describes you:

○ I work in a small firm (1–9 employees) (1)
○ I work in a midsize firm (10–49 employees) (2)
○ I work in a large firm (50 or more employees) (3)
○ I am self-employed (4)
○ Other (5) ________________________________________

Q5 The firm you work is within the area of:

○ Architecture (1)
○ Engineering (2)
○ Building performance consultancy (3)
○ Building surveying (4)
○ Multi-disciplinary (5)
○ Other: (6) ________________________________________

Q6 How long you have been performing Whole Building Life Cycle Assessment (WBLCA)?

○ Less than 1 year (1)
○ 1 to 5 years (2)
○ 5 to 10 years (3)
○ Over 10 years (4)

Q7 What statement best describes your level of understanding of WBLCA?

○ I know how to interpret some of the results of a WBLCA (1)
○ I know how to interpret the results of a WBLCA, but I need help making environmental improvements to the design (2)
○ I know how to interpret the results of a WBLCA, and I can make environmental improvements to the design accordingly (3)
○ I can help others to perform a WBLCA and make environmental improvements to the design (4)

Q8 What is your main motivation to perform a WBLCA?

○ To comply with a certification system, code or regulation. (1)
○ Client's request (2)
○ Educational purpose (3)
○ Other: (4) ________________________________________

Q9 To which certification system, code or regulation did you comply? Select all that apply.

○     LEED (1)
○     BREEAM (2)
○     Green Star (3)
○     HQE (4)
○     CASBEE (5)
○     DGNB (6)
○     ILFI—Living Building Challenge (7)
○     Green Globe (8)
○     CaGBC's Zero Carbon Building Standard (9)
○     ISCA (10)
○     CEEQUAL (11)
○     Country's specific code or regulation. Please specify below: (12) \_\_\_\_\_\_\_\_\_\_\_\_\_\_\_\_\_

Q10 What would motivate you to perform a WBLCA more frequently? Drag and drop the motivations presented below, ranking them in order of relevance (most relevant at the top).

Motivations to perform a WBLCA more frequently

\_\_\_\_\_\_ If WBLCA methodology was easier to perform (1)

\_\_\_\_\_\_ If WBLCA was included in my formal education, culture and practice (2)

\_\_\_\_\_\_ If there were more collaboration between designers and other WBLCA sectors, such as industry, researchers and tool developers (3)

\_\_\_\_\_\_ If there were more WBLCA demand, for example, from clients, certification bodies and government (4)

\_\_\_\_\_\_ If the time and cost to perform WBLCA were reduced (5)

\_\_\_\_\_\_ If WBLCA software tools were more aligned with the building design process (6)

\_\_\_\_\_\_ Other: (7)

Q11 In what stages of the building's project do you usually perform a WBLCA? Select all that apply.

○     Early stages (1)
○     Developed stages (2)
○     Handover stages (3)
○     Use stage (4)
○     I do not know (5)

Q12 Which software tool do you usually use to perform a Whole Building Life Cycle Assessment (WBLCA)?

If you have used more than one, please select the one you are most familiar with.

○     CAALA (1)
○     Etool LCD (2)
○     One Click LCA (3)
○     Athena (4)
○     Tally (5)
○     Elodie (6)
○     eco2soft (7)
○     eLCA (8)
○     BEES (9)
○     EcoEffect (10)
○     Pleiades ACV EQUER (11)
○     Other: (12) \_\_\_\_\_\_\_\_\_\_\_\_\_\_\_\_\_\_\_\_\_\_\_\_\_\_\_\_\_\_\_\_\_\_\_\_\_\_

Q13 Which of the following attributes does the software have? Select all that apply.

Show the reasoning behind assumptions and results (2)

○     Show graphical outputs (3)
○     Show numerical or table format outputs (4)
○     Indicates the range in which the results fall, instead of a single value (e.g., the GWP results fall within 100 and 150 Kg CO2eq) (5)
○     Provide a pre-set library with predefined WBLCA assumptions (1)

○ Provide an interface that adapts to the assessment scope (6)
○ Provide technical Support (7)
○ Save previous design solutions (8)
○ Compare results with other buildings or design alternatives (9)
○ Allow you to edit the assumptions in a pre-set library (14)
○ Comply with code or certification (10)
○ Suggest design alternative (e.g., alternative materials or another technical solution) (11)
○ Connect with 3D model (12)
○ Connect with other building performance tools (13)
○ Provide fast or instant results (15)
○ Combine LCA with cost analysis (16)
○ Offer a free license version of the software (17)

Q14 How useful are these attributes for the assessment?

| | | | | | |
|---|---|---|---|---|---|
| Show the reasoning behind assumptions and results (x2) | ☆ | ☆ | ☆ | ☆ | ☆ |
| Show graphical outputs (x3) | ☆ | ☆ | ☆ | ☆ | ☆ |
| Show numerical or table format outputs (x4) | ☆ | ☆ | ☆ | ☆ | ☆ |
| Indicates the range in which the results fall, instead of a single value (e.g., the GWP results fall within 100 and 150 Kg $CO_2$eq) (x5) | ☆ | ☆ | ☆ | ☆ | ☆ |
| Provide a pre-set library with predefined WBLCA assumptions (x1) | ☆ | ☆ | ☆ | ☆ | ☆ |
| Provide an interface that adapts to the assessment scope (x6) | ☆ | ☆ | ☆ | ☆ | ☆ |
| Provide technical Support (x7) | ☆ | ☆ | ☆ | ☆ | ☆ |
| Save previous design solutions (x8) | ☆ | ☆ | ☆ | ☆ | ☆ |
| Compare results with other buildings or design alternatives (x9) | ☆ | ☆ | ☆ | ☆ | ☆ |
| Allow you to edit the assumptions in a pre-set library (x14) | ☆ | ☆ | ☆ | ☆ | ☆ |

| Comply with code or certification (x10) | ☆ | ☆ | ☆ | ☆ | ☆ |
| Suggest design alternative (e.g., alternative materials or another technical solution) (x11) | ☆ | ☆ | ☆ | ☆ | ☆ |
| Connect with 3D model (x12) | ☆ | ☆ | ☆ | ☆ | ☆ |
| Connect with other building performance tools (x13) | ☆ | ☆ | ☆ | ☆ | ☆ |
| Provide fast or instant results (x15) | ☆ | ☆ | ☆ | ☆ | ☆ |
| Combine LCA with cost analysis (x16) | ☆ | ☆ | ☆ | ☆ | ☆ |
| Offer a free license version of the software (x17) | ☆ | ☆ | ☆ | ☆ | ☆ |

Q15 Which additional attributes do you wish the software had?

○ Other: (1) ________________________________________________
○ Show the reasoning behind assumptions and results (3)
○ Show graphical outputs (4)
○ Show numerical or table format outputs (5)
○ Indicates the range in which the results fall, instead of a single value (e.g., the GWP results fall within 100 and 150 Kg CO2eq) (6)
○ Provide a pre-set library with predefined WBLCA assumptions (2)
○ Provide an interface that adapts to the assessment scope (7)
○ Provide technical Support (8)
○ Save previous design solutions (9)
○ Compare results with other buildings or design alternatives (10)
○ Allow you to edit the assumptions in a pre-set library (15)
○ Comply with code or certification (11)
○ Suggest design alternative (e.g., alternative materials or another technical solution) (12)
○ Connect with 3D model (13)
○ Connect with other building performance tools (14)
○ Provide fast or instant results (16)
○ Combine LCA with cost analysis (17)
○ Offer a free license version of the software (18)

Q16 What type of outputs would help you to make more informed decisions?
The graphical examples are intended to provide a better understanding of the alternatives. They are not actual assessment results.

Q17 Regarding building life cycle stages:
*Rank the following options in order of preference—1 being your favourite and 3 being your least favourite.*

______ Embodied vs. Operational (1)

_______ Grouped life cycle stages (2)

_______ All life cycle stages (3)

Q18 Regarding building materials:

Rank the following options in order of preference—1 being your favourite and 3 being your least favourite.

_______ Categorised by construction systems (1)

_______ Grouped in main materials (2)

_______ List of more detailed materials (3)

Q19 Regarding environmental impact categories:

Rank the following options in order of preference—1 being your favourite and 3 being your least favourite.

_______ Show only one impact category, such as Global Warming Potential (GWP) (1)

_______ Show Endpoint impact categories (2)

_______ Show Midpoint impact categories (3)

Q20 Regarding the precision of outputs:

Rank the following options in order of preference—1 being your favourite and 3 being your least favourite.

_______ Show the level of performance (1)

_______ Show the output range (2)

_______ Show precise outputs (3)

Q21 Regarding the functional unit:

Rank the following options in order of preference—1 being your favourite and 4 being your least favourite.

_______ Whole building (1)

_______ Per unit (e.g., per dwelling if residential or per office if commercial building) (2)

_______ Per metre square (3)

_______ Per occupant (4)

Q22 Regarding comparison of results:

Rank the following options in order of preference—1 being your favourite and 4 being your least favourite.

_______ I would prefer not to compare results (1)

_______ I would prefer to compare my design with a benchmark (2)

_______ I would prefer to compare different design alternatives of the same building (3)

_______ I would prefer to compare my design with other similar buildings (4)

Q23 Regarding the type of graphical representation

Rank the following options in order of preference—1 being your favourite and 4 being your least favourite.

_______ Pie (1)

_______ Bars (2)

_______ Columns (3)

_______ Radar (4)

Q24 Regarding other output options. Select all that apply

○ See all environmental impacts in the same graph as per the radar graphs (1)

○ Have a single environmental impact index representing all environmental impacts (2)

○ Be flexible on the type of graph and the level of output information (e.g., having the option of seeing the materials categorised in construction systems or see a more detailed list of materials) (3)

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
