# Peer review of "International Survey of the Usage of Whole Building Life Cycle Assessment Software"

_buildings, doi:10.3390/buildings12081278_

Round 1

Reviewer 1 Report

The subject matter described in the article relates to important issues in the process of building design, supported by IT tools.                                            Please refer at work by the authors to the following information:                   

A lot of experience is needed to be a specialist in a given field. The survey, which is the basis for the analyzes, shows that over 70% of respondents perform WBLCA less than 5 years. In such a situation, can the analyzes carried out and the final conclusions drawn be generalized to all users? Should inference in the discussed scope not refer only to people who are beginners in the subject of WBLCA implementation? It is suggested to introduce the above provisions in the presented work.

Additionally, it seems that the authors could, on the basis of the performed analyzes, indicate in their conclusions more bold suggestions for software developers in the described subject, as it was assumed for the purpose of the study.

Author Response

We appreciate your time in reviewing our manuscript. Your comments are thoroughly addressed and are greatly appreciated.

Please see the attachment for the point-by-point response.

Reviewer 2 Report

This manuscript addresses an interesting issue regarding the application of life-cycle assessment in building design. This is an important topic in the international agenda, as the building sector represents an important share in terms of final energy consumption and GHG emissions. The goals to be addressed are adequately framed within previous research conducted by the authors, and the topics are presented in a transparent and comprehensive way. A survey to building designers using LCA tools in the design process was conducted, with very interesting and statistically relevant outcomes. Recommendations are made that may potentially improve the development of more efficient and design-oriented tools. Finally, the manuscript is well-structured and includes a comprehensive list of references.

Author Response

We appreciate your comments and time in reviewing our manuscript. 

Reviewer 3 Report

From the year 2000, when approximately a dozen articles were published, to the nearly 800 that have been published in recent years on "Life Cycle Assessment", the great importance that the issue is acquiring is evident. That is why the interest of the revised document is initially recognized in which it is analyzed through a survey of building designers, engineers and architects, who use WBLCA software simulations as a tool for decision making in this area, trying to identify patterns in the use of software capabilities to provide guidance to WBLCA software developers, improve the experience and efficiency of the tool.

However, there is a series of issues that must be taken into account and that influence the decision to accept the document, which are explained below:

Title (lines 2-3): really the title that responds to the content would be "Survey the usage of the software of Whole Building Life Cycle Assessment"

Abstract (lines 10-24): Based on the content, it is considered correct.

 Keywords (lines 26-26) replace “Software tools” with “Survey the usage of the software”

1 Introduction (lines 28 to 103): in principle it can be considered correct.

2. Materials and Methods (lines 104-175): The survey is the basic tool of this research and it is observed that the basic or essential data for its conformation are provided (online survey, June and December of 2021, targeting building designers who use WBLCA software tools for decision-making, published on a professional platform, it was advertised in many groups related to the environmental impact of buildings, contacted individually via the same platform and e-mail, online survey was also published in the Carbon Leadership Forum community in its August 2021, etc).

Also in Figure 1 (line 117), have the structured in three parts: (1) demographic questions, (2) software attributes and (3) informative outputs. On the other hand, the Likert scale was used to 119 rate software attributes in which 1 star represented “not useful at all” and 5 stars repre-120 sented “extremely useful”. In Table 1 (line 133), the attributes were categorized into 4 groups, design-oriented, inter-disciplinary connection, data-transparency, and motivational. In Table 2 (line 134), the informative outputs were categorized as building life cycle stages, building materials, environmental impact categories, precision of outputs, functional unit, comparison of results and type of graphical representation. The statistical tests are summarized in Table 3. Statistical tests performed in each step are in table 4.

Given the explanations and the documents provided, it is understood that the requirements of a survey of this type are basically covered and that the survey has already been carried out, so there is little room for change or improvement, but certain general issues that must be taken into account regarding said survey and the inclusion, where appropriate, of the appropriate references (Salvador-Oliván J.A. et al 2021: Evaluating survey research in articles published in Library Science journals/Spanish Journal of Scientific Documentation 44 (2)):

Although survey research is a frequently used method and the survey has been defined as a systematic method of data collection, conducting proper survey research is a "science" that requires the same planning and structure as any other study that employs survey research. the quantitative method, where it is necessary to define mainly (Morgan and Carcioppolo, 2014): a) the population to study, number of individuals to include and how they will be selected; b) what data will be collected (questionnaire design); c) how the questionnaire will be distributed (administration method); and d) when to start the survey and follow up on non-responders. The replicability of a study is a cornerstone of the scientific method, and to ensure that it is possible, transparency must be provided in the information on the sufficient work carried out (McNutt, 2014). It is the obligation of the researcher to describe the methods and find them accurately and in detail, complying with the standards (American Association for Public Opinion Research (AAPOR), 2015a). To promote high-quality data (Logan et al., 2020), errors must be avoided (Jedinger et al., 2018), which can be classified into two large groups: those due to observation or measurement and errors due to non-observation. The three types of errors due to non-observation are: (1) coverage error, (2) non-response error, and (3) scanning error. And the possible sources of observation errors: (1) the interviewer, (2) the respondent, (3) the questionnaire, and (4) the mode of data collection (Groves, 1987; Weisberg, 2018). Observation and non-observation errors make up the total error that can be made in a survey (Groves and Lyberg, 2010). Without clear and complete information on the procedures on which the surveys are based, it is difficult to be able to assess the possible errors and the quality of the research, hence the importance of researchers providing complete and exhaustive information on the design has been highlighted of the survey, as well as on the process and statistical analysis of the data collected. The quality of survey data remains a primary concern, and the quality of the methodology used in survey studies is essential both to ensure the reliability and validity of the results and to generalize the results to the populations from which the samples are drawn analyzed.

The lack of information compromises both the transparency and the trustworthiness of the research (Bennett et al., 2011). 

Recently, recommendations have been published to design and create the best possible survey (AAPOR, 2020), complemented by a series of questions that help evaluate the validity and quality of the results of a survey (Baker et al., 2016 ) ). Other recommendations of interest on best practices in survey research are those published in a report by the National Science Foundation based on two conferences held (Krosnick, et al., 2015).

Guidelines have also been developed (Wharton, 2017) that indicate the criteria that authors must report so that readers can critically assess the quality of the research (Hui et al., 2019). The highest quality survey sometimes does not comply with survey design and data collection manuals (Dale, 2006) (Totten et al., 1999). In these cases, there is a common denominator: enough information to assess the quality of the surveys.

3. Results and discussion (lines 176-415): in general they are of interest and have been developed, however the following suggestions are made:

3.1. Relationship between groups of participants (lines 181-213)

It is understood that the content is correct, but perhaps the title of this section is not so correct with respect to said content and could be qualified.

3.2. Factor of attributes you want to have (WIH) (lines 215-275)

Some statements about comments on the results must be nuanced or expanded by applying principles of contradiction, methodical doubt, etc.

3.3. Attribute utility rating (lines 276-335)

As in the previous section, some statements about comments on the results must be nuanced or expanded by applying principles of contradiction, methodical doubt, etc.

3.4. Output preferences (lines 337-414)

It seems to be adequate, but also take into account the previous comments

4. Conclusion (lines 417-473): It is understood that they are mostly too generic and perhaps they could be reduced a little more in those general aspects already reiterated, especially when there are tables like 9 that summarize the situation very well. It is recommended that they adapt to everything mentioned above and that they focus on what is truly interesting and novel.

Appendix A – Online survey: It has already been done, so it is not possible to modify it based on what is indicated in the methodological section.

5 The references (lines 770)-…: some of them should be updated and those that complete what was indicated in previous comments should be added.

Finally, it is understood that it would be of great interest to include a section, before the conclusions, with the limitations of the study, and a special mention of the trends and evolution of the software and how this can influence the results.

Kinds regards

Author Response

We appreciate your comments and time in reviewing our manuscript. Your comments were carefully addressed, contributing significantly for the improvement of the manuscript.

Please see the attachment for the point-by-point response.

Round 2

Reviewer 3 Report

Dear Authors,

Some of the aspects indicated in the first review have been modified and although there are some points that could be improved and a certain fragility persists in the survey tool used, due to the interest in the revised document it is understood that it could be published.

Kindly regards